# Morphometric and Nutritional Characterization of the Main Spanish Lentil Cultivars

**Javier Plaza** [1,*] 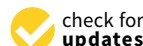, **M. Remedios Morales-Corts** [1] , **Rodrigo Pérez-Sánchez** [1] , **Isabel Revilla** [2]
and **Ana M. Vivar-Quintana** [2]

1 Plant Production Group, Faculty of Environmental and Agricultural Sciences, University of Salamanca, 37007 Salamanca, Spain; reme@usal.es (M.R.M.-C.); rodrigopere@usal.es (R.P.-S.)
2 Food Technology Group, Superior Polytechnic School of Zamora, University of Salamanca, 49022 Zamora, Spain; irevilla@usal.es (I.R.); avivar@usal.es (A.M.V.-Q.)
* Correspondence: pmjavier@usal.es; Tel.: +34-654-518-435

**Abstract:** Nowadays, there is a growing demand for high-quality vegetal protein food products, such as pulses and lentils in particular. However, there is no scientific evidence on the nutritional and morphometric characterization of the main lentil cultivars in the Western Mediterranean area. For this reason, the aim of this work is to carry out a morphometric and nutritional characterization of the main Spanish lentil cultivars. Nutrient content assessment was performed on dry matter. The results showed that all studied cultivars are large and heavy lentils, except for the cultivar "Pardina". They have high protein levels, ranging from 21% to 25%, which is higher than those found in any other pulse, as well as a high carbohydrate content, greater than 59% in all cases. Fiber content was higher than expected in "Armuña" and "Rubia Castellana" cultivars, ranging from 6% to 6.6%, and exceptionally high in the case of the cultivar "Pardina", which reached 7.8%. Conversely, very low values were found for fat content, varying between 0.5% and 0.9%. Ca, Fe and Mg levels were remarkably higher (from 550 ppm to 851 ppm, from 98 ppm to 139 ppm and from 790 ppm to 989 ppm, respectively) than those found for other lentil cultivars, especially the high Mg content in the cultivars "Jaspeada" and "Microjaspeada", both above 955 ppm. Clear differentiation was found between the cultivars "Rubia Castellana", "Pardina" and those included in the Protected Geographical Indication (PGI) "Lenteja de la Armuña". Overall, lentil cultivars included in the PGI "Lenteja de la Armuña" showed better morphometric and nutritional characteristics than cultivars "Pardina" or "Rubia Castellana".

**Keywords:** *Lens culinaris*; morphological characteristics; chemical composition; quality mark; protein intake

## 1. Introduction

In recent years, Europe has undergone a marked change in the dietary habits of its population, especially in Mediterranean countries, including Spain. This change is driven by the growing consumer demand for vegan or vegetarian food products [1], leading the agricultural sector to produce more vegetable-based foods, focusing on pulses due to their high protein content [2]. Furthermore, the European Union has recently committed itself to reduce greenhouse gas emissions from the food sector by increasing the production of vegetable protein rather than animal protein [3–5].

Pulses are one of the food sources with the best nutritional properties whose consumption is associated with several benefits for human health [6,7]. The inverse relationship between pulses consumption and the risk of suffering from coronary heart disease, type II diabetes and obesity is particularly noteworthy. Moreover, pulses are associated with low levels of serum LDL cholesterol and high levels of serum HDL cholesterol [8,9]. Besides, legumes are essential components in sustainable agroecosystems as they are able to carry out biological nitrogen fixation [10,11]. Consequently, not only do they not require nitrogen

supplementation, but they also significantly reduce the amount of nitrogen that must be added to subsequent crops. Therefore, these properties make pulses a very powerful tool for tackling malnutrition (especially protein malnutrition) and for reducing fossil fuel and chemical fertilizer consumption in traditional farming systems [12].

Among the broad range of species constituting the *Fabaceae* family, lentils (*Lens culinaris* Med.) stand out as one of the most important sources of protein, mainly in developing countries due to their low production cost [5], and they are also considered a staple food in the Mediterranean diet [13]. Besides being very rich in protein and low in fat, lentils provide necessary amino acids such as lysine, even though they have low concentrations of methionine and cysteine [14]. Lentils have high content of complex carbohydrates, especially starch. Thus, they are also an important source of energy. Moreover, lentils are high in fiber [15], water-soluble vitamins, essential minerals and numerous phenolic compounds such as tannins, which are correlated with high in vitro antioxidant capacity [16]. Lentils are considered as soft seed-coated pulses requiring a shorter cooking time, reason why the usually have smaller nutrient losses than those with a hard seed coat [17]. Despite the nutritional benefit and the contribution to the environmental sustainability, the difficulties in mechanization of some agronomic practices combined with the low yields due to adverse environmental conditions, such as salinity [18], and weak competitiveness against weeds [19,20], make lentils a crop of low productivity which restricts its expansion in Mediterranean countries [5].

Despite these limitations, according to the Agricultural Statistics Annual Report prepared by the Ministry of Agriculture, Fisheries and Food (MAPA), 44,100 hectares of lentils were cultivated in Spain in 2019, which resulted in 42,800 tons of production, almost twice as much as in the 2018 campaign. In Spain, a significant amount of lentil production is included in a Protected Geographical Indication (PGI), a quality mark regulated by European Council Regulation (EC) 2081/92 that certifies that the lentil is associated with a specific geographical area and that it has been produced following established quality standards. The only two Spanish PGIs are the PGI "Lenteja de la Armuña" (EU 1151/2012), specific to the "Armuña" area which is part of the province of Salamanca (Castilla y León, Spain), and the PGI "Lenteja de Tierra de Campos", from the provinces of Valladolid and Palencia (Castilla y León, Spain), the former being highly appreciated for their nutritional value [21]. Within PGI "Lenteja de la Armuña", the cultivar "Rubia de la Armuña" is considered the precursor ecotype from which the rest of the cultivars admitted in this PGI have been gradually selected, being the main one the variety "Guareña". In turn, by applying size and color criteria, two new varieties were selected from "Guareña", named "Jaspeada" and "Microjaspeada". However, there are other lentil cultivars of great importance in Spain that are not included in a PGI, such as the "Rubia Castellana" lentil, from the region of Castilla La Mancha. In this sense, it would be interesting to assess the morphometric and nutritional characteristics of the different cultivars grown in the same experimental place.

Currently, very few scientific studies exist on the morphometric characterization and nutritional composition of these Spanish lentil cultivars [22,23]. However, some authors like Sonnante and Pignone [24], Scippa et al. [25] and Zaccardelli et al. [26] have conducted characterization works on Italian lentil landraces such as "Altamura", "Villalba" or "Linosa". Other Italian authors like Bacchi et al. [27] studied the morphologic traits and agronomic performances of 22 selected landraces from lentil germplasm collection maintained at Plant Genetics Institute (National Research Council-CNR-Bari), while Laghetti et al. [28] reviewed the lentil landraces traditionally cultivated in Italy, such us "Colfiorito", "Onano", "Ventotene", "Ustica", "Pantelleria" or the three mentioned cultivars studied by Scippa et al. [25], among others. Similarly, Toklu et al. [29] performed agromorphological studies of Turkish lentil cultivars such as "Firat 87", "Yerli Kirmizi" or "Emre", among others, and Idrissi et al. [30] used 19 Simple Sequence Repeat DNA markers for molecular variance analysis (AMOVA) and population structure assessment underlying 74 lentil landraces from Morocco, Italy, Greece and Turkey.

Considering the current trend of increasing consumption of pulses like lentils, it becomes necessary to establish food quality programs [31] that delve into the relevant nutritional compounds of Spanish lentils, as well as their main morphometric characteristics. Therefore, the aim of the present study is to carry out a morphometric and nutritional characterization of the main Spanish lentil cultivars.

## 2. Materials and Methods

### 2.1. Experimental Design and Plant Material

Two growing seasons from October to July 2017–2018 (2017) and 2018–2019 (2018) were conducted in the experimental fields of the University of Salamanca (40°56′44″ N, 5°39′40″ W, 795 m above sea level (a.s.l.)). The edaphoclimatic conditions of the experimental plots are associated with a continental Mediterranean climate (Table 1) and sandy-clay-loam basic soils (Table 2), optimal for lentil cultivation due to its low tolerance to acidic soils.

**Table 1.** Meteorological conditions from October to July for 2017–2018 (2017) and 2018–2019 (2018) growing seasons in the central plateau of the Iberian Peninsula.

| Parameter | Period | |
|:---:|:---:|:---:|
| | 1 October 2017–10 July 2018 | 1 October 2018–10 July 2019 |
| Average temperature (°C) | 8.99 | 9.28 |
| Total rainfall (mm) | 338.01 | 257.53 |
| Total sunshine radiation (MJ/m$^2$) | 3923.84 | 3929.19 |

Data provided by Spanish National Meteorological Agency (AEMET).

**Table 2.** Analytical soil characteristics of the experimental plots.

| Parameter | Value |
|:---:|:---:|
| Texture | Sandy clay loam |
| pH | 7.8 |
| $P_2O_5$ (ppm) | 122 |
| $K_2O$ (ppm) | 32 |
| Organic Matter (%) | 2.3 |
| Class | Calcic Cambisol |

In each of the growing seasons, 44 plots ($5 \times 25$ m$^2$ each one) were used to grow six different Spanish lentil cultivars (Figure 1). Among these plots, 36 belonged to lentil cultivars included in the PGI "Lenteja de la Armuña", from which nine plots belonged to cultivar "Rubia de la Armuña", nine to cultivar "Guareña", nine belonged to cultivar "Jaspeada" and nine to cultivar "Microjaspeada". Among the remaining eight plots, four belonged to the cultivar "Pardina", included in the PGI "Lenteja de Tierra de Campos" and the last four plots corresponded to the cultivar "Rubia Castellana", from the province of Cuenca (Castilla La Mancha, Spain). All the samples used in this study belong to the species *Lens culinaris* subsp. *macrosperma*, except for the cultivar "Pardina" which belongs to subsp. *microsperma*. All this information is presented in Table 3.

**Table 3.** Identification of the studied cultivar and number of experimental plots used.

| PGI | Cultivar | Subsp. | No. Plots |
|:---:|:---:|:---:|:---:|
| "Lenteja de la Armuña" | "Guareña" | *Macrosperma* | 9 |
| "Lenteja de la Armuña" | "Rubia de la Armuña" | *Macrosperma* | 9 |
| "Lenteja de la Armuña" | "Jaspeada" | *Macrosperma* | 9 |
| "Lenteja de la Armuña" | "Microjaspeada" | *Macrosperma* | 9 |
| "Tierra de Campos" | "Pardina" | *Microsperma* | 4 |
| - | "Rubia Castellana" | *Macrosperma* | 4 |

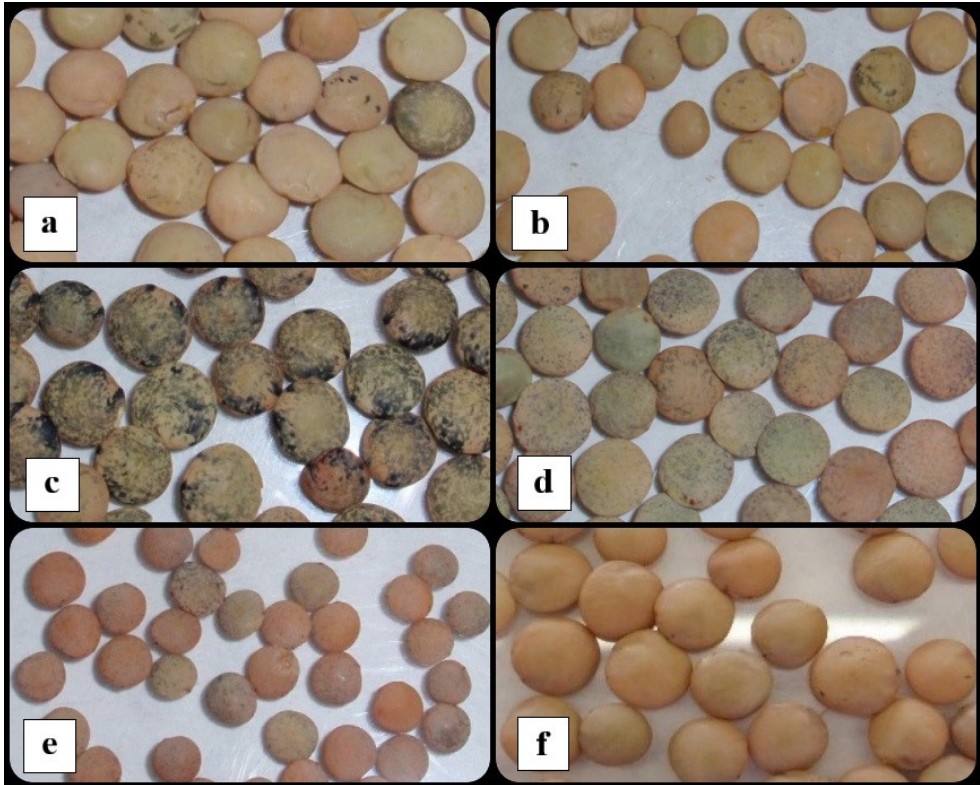

**Figure 1.** The lentil cultivars used in this work: (**a**) Guareña; (**b**) Rubia de la Armuña; (**c**) Jaspeada; (**d**) Microjaspeada; (**e**) Pardina; (**f**) Rubia Castellana.

Regarding growing practices, the same field preparation work was carried out for all plots, in which neither fertilizers nor pesticides were used. A sowing rate of 100 kg of seed per hectare was used on all plots. At harvest time, a one kg seed sample was collected from each plot and used for morphometric and nutritional characterization.

### 2.2. Morphometric Seed Characteristics

The morphometric characteristics assessed in this study were: shape of the longitudinal section, main color, secondary color pattern, distribution of the secondary color pattern, size and weight.

All the morphometric parameters mentioned were measured mainly following the descriptors established by the International Union for the Protection of New Varieties of Plants (UPOV) [32]. The distribution of the secondary color pattern on the lentil surface (%) was estimated by dividing the lentil into four quadrants in a Petri dish, and then relating the number of mottles in each quadrant by its surface area and averaging the four quadrants results to obtain the secondary pattern's final percentage of coverage. Lentil size refers to the longitudinal diameter expressed in millimeters and a digital Vernier caliper was used for its measurement. For these parameters, 70 lentils per sample were evaluated. As for the longitudinal diameter, three measurements for the same lentil sample were performed. The mean of these three measurements was calculated for each of the 70 lentils of each cultivar assessed. Subsequently, those 70 mean results were averaged for each cultivar. Weight was calculated, after drying the sample, as the average weight of 100 seeds using a precision balance (three replications per cultivar were measured).

### 2.3. Nutritional Seed Quality

Nutritional quality was quantified by analyzing the following parameters: moisture, Ca, Fe and Mg content (as representatives of the main mineral elements), ash, crude protein, total fat, total fiber and carbohydrates.

The samples were ground with their skin in Foss Knifetec<sup>TM</sup> 1095 mill (FOSS, Hilleroed, Denmark), with temperature control at 15 °C. For moisture calculations, samples were dried following the Association of Official Analytical Collaboration (AOAC) International procedures [33], i.e., samples were dried in a conventional oven at 100 °C during 3 h until constant weight. The ash content was determined by quantifying the residue after combustion of the dry sample in a muffle furnace at 540 °C for 6 h under conditions corresponding to the gravimetric method. Both parameters were measured according to the AOAC standards [33]. The mineral elements Ca, Fe and Mg were analyzed by a Jobin Yvon Ultima II inductively coupled plasma source atomic emission spectrometer (ICP-OES) (Horiba Jobin Yvon, Lille, France). Previously, the samples were disaggregated in a pressure reactor with nitric acid. For the calculation of crude protein, the nitrogen content was previously determined by the Kjeldahl method and, subsequently, a correction factor of 6.25 was applied to estimate the protein content [33]. Total fat content was determined by the Soxhlet method of extraction with petroleum ether, according to standard AOAC [33]. Total fiber content was analyzed following the method proposed by Goering and Van Soest [34] using ANKOM equipment (Ankom Technology, Macedon, NY, USA). The carbohydrate content was estimated by differences between the dry extract (obtained after oven drying) and the rest of the components. All determinations were performed on dry matter in triplicate.

*2.4. Statistical Analysis*

Statistical processing of the data was carried out using IBM-SPSS Statistics 26 software (IBM, Chicago, IL, USA). Significant differences among cultivars for each of the morphometric and nutritional quality parameters were obtained by multivariate analysis of variance (MANOVA) fitted to a general linear model (GLM). Means and standard deviations (SD) were calculated for all variables. The statistical significance of each factor was assessed at a 95% confidence level ($\alpha = 0.05$) using Snedecor's F as the contrast statistic. For differentiation of homogeneous subsets, Tukey's test was used. This procedure was also used to study significant differences between growing seasons.

To represent graphically in a two-dimensional space the separation between the different Spanish cultivars measured in terms of Mahalanobis distance, a stepwise discriminant analysis was performed. This technique was carried out using only the variables that refer to the nutritional quality of the samples. Thus, the analysis was not biased by the great variability in morphometric parameters, which would completely distort the differences caused by nutritional parameters. Furthermore, it can be asserted that the differences found between the positions of the samples in the two-dimensional space are exclusively due to the inherent nutritional quality of each of the lentil cultivars involved in this work.

In order to verify the stepwise discriminant results, a cluster analysis was performed following an agglomerative hierarchical algorithm applying Ward's method to maximize intra-cluster homogeneity. All variables were used, after standardized them to Z scores.

**3. Results**

Table 4 shows the shape type, the main color, the secondary color pattern and the means and standard deviations of the rest of the morphometric characters and Table 5 shows the means and standard deviations of each of the nutritional parameters.

It should be noted that the results shown below correspond to the average data of the two growing seasons (2017 and 2018). In the studies dealing with plant material characterization, it is essential to assess different growing seasons to take into account climatic conditions influences, but the results are usually expressed as the average data of the seasons involved, as in the case of the study carried out by Bacchi et al. [27]. However, the comparative study between growing seasons was carried out and the synthetic results are shown in Table 6.

**Table 4.** Morphometric characteristics of the different lentil varieties (mean ± SD).

| Parameter | Guareña (*n* = 9) | Rubia de la Armuña (*n* = 9) | Microjaspeada (*n* = 9) | Jaspeada (*n* = 9) | Pardina (*n* = 4) | Rubia Castellana (*n* = 4) | Mean (N = 44) |
|---|---|---|---|---|---|---|---|
| Form | Medium elliptical | Medium elliptical | Medium elliptical | Medium elliptical | Wide elliptical | Medium elliptical | - |
| Main color | Green-Pink | Pink-Green | Green-Pink | Green-Pink | Pink-Green | Pink | - |
| Secondary color pattern | Blotched | Blotched | Spotted | Spotted-marbled | Spotted | Spotted | - |
| Secondary color pattern distribution (%) | 16.4 ± 1.3 [c] | 14.7 ± 4.4 [c] | 96.7 ± 0.5 [a] | 98.5 ± 1.1 [a] | 75.4 ± 4.2 [b] | 5.8 ± 0.5 [d] | 53.7 ± 41.0 |
| Weight (g) | 6.85 ± 0.43 [ab] | 6.33 ± 0.45 [bc] | 7.07 ± 0.08 [a] | 7.32 ± 0.34 [a] | 3.58 ± 0.24 [d] | 5.91 ± 0.15 [c] | 6.50 ± 1.07 |
| Diameter (mm) | 6.87 ± 0.27 [a] | 6.58 ± 0.30 [a] | 6.55 ± 0.11 [a] | 6.64 ± 0.16 [a] | 4.43 ± 0.09 [c] | 5.86 ± 0.09 [b] | 6.39 ± 0.70 |

[abcd] Different letters mean statistically significant differences $p < 0.05$. Statistical differences are to be compared within rows. SD: standard deviation, n: number of plots.

**Table 5.** Nutritional characteristics of the different lentil varieties (mean ± SD).

| Parameter | Guareña (*n* = 9) | Rubia de la Armuña (*n* = 9) | Microjaspeada (*n* = 9) | Jaspeada (*n* = 9) | Pardina (*n* = 4) | Rubia Castellana (*n* = 4) | Mean (N = 44) |
|---|---|---|---|---|---|---|---|
| Moisture (%) | 6.47 ± 0.45 [a] | 6.58 ± 0.71 [a] | 5.45 ± 0.44 [b] | 5.48 ± 0.51 [b] | 5.79 ± 0.64 [ab] | 5.46 ± 0.28 [b] | 5.93 ± 0.72 |
| Protein (%) | 24.6 ± 1.3 [a] | 24.6 ± 1.9 [a] | 21.8 ± 0.8 [b] | 21.2 ± 0.6 [b] | 22.4 ± 0.7 [b] | 25.0 ± 0.1 [a] | 23.2 ± 1.9 |
| Carbohydrates (%) | 60.2 ± 2.4 [bc] | 59.9 ± 2.4 [c] | 63.1 ± 0.9 [ab] | 63.4 ± 0.9 [a] | 61.5 ± 1.6 [abc] | 60.6 ± 0.4 [abc] | 61.6 ± 2.2 |
| Total Fiber (%) | 6.6 ± 0.3 [b] | 6.2 ± 0.3 [bc] | 6.1 ± 0.1 [c] | 6.2 ± 0.1 [bc] | 7.8 ± 0.1 [a] | 6.2 ± 0.2 [c] | 6.4 ± 0.5 |
| Ashes (%) | 2.3 ± 0.1 | 2.2 ± 0.2 | 2.4 ± 0.1 | 2.3 ± 0.1 | 2.3 ± 0.1 | 2.5 ± 0.2 | 2.3 ± 0.2 |
| Total fat (%) | 0.7 ± 0.0 [b] | 0.6 ± 0.0 [bc] | 0.8 ± 0.0 [a] | 0.7 ± 0.1 [b] | 0.9 ± 0.1 [a] | 0.5 ± 0.1 [c] | 0.7 ± 0.1 |
| Ca (ppm) | 792.3 ± 69.0 [ab] | 786.1 ± 108.9 [ab] | 850.8 ± 28.6 [a] | 710.5 ± 14.7 [b] | 809.2 ± 16.5 [ab] | 550.2 ± 1.8 [c] | 765.8 ± 100.9 |
| Fe (ppm) | 138.9 ± 17.0 [a] | 130.4 ± 31.5 [ab] | 124.7 ± 5.0 [abc] | 127.3 ± 4.0 [ab] | 98.5 ± 8.2 [c] | 106.3 ± 4.4 [bc] | 125.3 ± 20.0 |
| Mg (ppm) | 913.4 ± 47.6 [a] | 952.9 ± 46.7 [a] | 988.0 ± 58.9 [a] | 957.81 ± 6.0 [a] | 805.3 ± 69.9 [b] | 790.4 ± 9.2 [b] | 924.8 ± 78.1 |

[abc] Different letters mean statistically significant differences $p < 0.05$. Statistical differences are to be compared within rows. SD: standard deviation, n: number of plots.

**Table 6.** Growing season's differences in the morphometric and nutritional parameters of the studied cultivars.

| Cultivar | Morphometric Parameters | | | | | | Nutritional Parameters | | | | | | | | |
|---|---|---|---|---|---|---|---|---|---|---|---|---|---|---|---|
| | FO | MC | SCP | SCPD | WE | DI | MO | PR | CH | TFi | AS | TFa | Ca | Fe | Mg |
| Guareña | ND | ND | ND | ND | * | * | ND | * | * | * | ND | ND | * | ND | ND |
| Rubia de la Armuña | ND | ND | ND | ND | * | * | ND | * | * | * | ND | ND | ND | ND | ND |
| Microjaspeada | ND | ND | ND | ND | * | * | ND | * | * | * | ND | ND | ND | ND | * |
| Jaspeada | ND | ND | ND | ND | * | * | ND | * | * | * | ND | ND | ND | ND | ND |
| Pardina | ND | ND | ND | ND | * | * | ND | * | * | * | ND | ND | ND | * | ND |
| Rubia Castellana | ND | ND | ND | ND | * | * | ND | * | * | * | * | ND | ND | ND | ND |

* Significant differences in the row $p < 0.05$, ND: no significant differences in the row $p > 0.05$. FO: form, MC: main color, SCP: secondary color pattern, SCPD: secondary color pattern distribution (%), WE: weight (g), DI: diameter (mm), MO: moisture (%), PR: protein (%), CH: carbohydrates (%), TFi: total fiber (%), AS: ashes (%), TFa: total fat (%), Ca: calcium (ppm), Fe: iron (ppm), Mg: magnesium (ppm).

### 3.1. Morphometric Seed Characteristics

All cultivars showed a medium elliptic shape except for the cultivar "Pardina" which was classified as wide elliptic. Considering the different determinations made on color, regarding the primary color, the cultivars "Guareña", "Microjaspeada" and "Jaspeada" showed a green-pink color, the cultivars "Rubia de la Armuña" and "Pardina" were cataloged as pink-green lentils and the cultivar "Rubia Castellana" presented a pink main color. As for the secondary color pattern, the cultivars "Guareña" and "Rubia de la Armuña" showed a blotched pattern, the cultivar "Jaspeada" a marbled-spotted pattern and the cultivars "Microjaspeada", "Pardina" and "Rubia Castellana" presented a spotted pattern. Regarding the secondary color pattern distribution, the cultivars "Jaspeada" and "Microjaspeada" showed a significantly higher surface distribution than the rest of the cultivars, being the cultivar "Rubia Castellana" the one with the least presence values.

Both weight and longitudinal diameter showed a wide range of variation due to the inclusion of *macrosperma* and *microsperma* morphotypes with statistically significant differences. For both variables, the cultivar "Pardina", since it is the only *microsperma* morphotype, had the lowest values, while the cultivar "Jaspeada" presented the highest weight and the cultivar "Guareña" had the highest diameter.

### 3.2. Nutritional Seed Quality

For moisture, statistically significant differences were found, being the cultivars "Guareña" and "Rubia de la Armuña" the ones that have higher moisture content.

Regarding protein content, the cultivars "Guareña", "Rubia de la Armuña" and "Rubia Castellana" showed statistically higher values (>24%) than cultivars "Microjaspeada", "Jaspeada" and "Pardina", ranging from 21% to 22.5%. For calculated carbohydrate content, statistically significant differences were observed among the different cultivars. The values obtained ranged between 59% and 63.5%, with the highest values for the cultivars "Jaspeada" and "Microjaspeada", while the lowest one was found for the cultivar "Rubia de la Armuña".

Total fiber content was significantly higher in the cultivar "Pardina" than in the rest of the cultivars. However, the differences found in the total ash content were not statistically significant. Regarding total fat content, it was again the cultivar "Pardina" who showed a significantly higher content than the rest, while the cultivar "Rubia Castellana" had the lowest total fat.

Statistically significant differences were observed in all of the mineral elements. Regarding Ca, all cultivars had a similar amount except for the cultivar "Rubia Castellana" which showed a significantly lower Ca level. Besides, it was the cultivar "Pardina" that presented a significantly lower amount of Fe, while the PGI "Lenteja de la Armuña" cultivars had the highest levels of this element, standing out the cultivar "Guareña" over the rest. As for Mg, the highest values were found in all PGI "Lenteja de la Armuña" cultivars, particularly high the cultivars "Jaspeada" and "Microjaspeada" Mg content.

### 3.3. Climatic Conditions Influence on the Studied Parameters

As shown in Table 1, 2018 growing season was hotter and considerably drier than the 2017 season. Overall, this variation significantly ($p < 0.05$) affected seed size and weight, as well as protein, carbohydrate and fiber content. These results are shown in Table 6.

### 3.4. Stepwise Discriminant and Cluster Analysis

Table 7 shows the structure matrix extracted from the stepwise discriminant analysis. It shows that total fiber, total fat, Mg and Ca were the variables that most influenced the distribution of the different samples in the two-dimensional space, whereas ashes and protein were excluded from the analysis.

**Table 7.** Structure matrix resulting from the stepwise discriminant analysis which shows correlations between variables and discriminant factors.

| Parameter | Function 1 (64.5%) | Function 2 (27.7%) |
|---|---|---|
| Total Fibre (%) | 0.577 * | −0.445 |
| Ashes (%) [b] | −0.085 | −0.277 |
| Ca (ppm) | 0.160 | 0.480 * |
| Mg (ppm) | −0.174 | 0.632 * |
| Carbohydrates (%) | 0.063 | 0.269 |
| Total fat (%) | 0.500 * | 0.470 |
| Protein (%) [b] | −0.089 | −0.335 |
| Fe (ppm) [b] | −0.166 | −0.076 |

* Largest absolute correlation between each variable and any discriminant function. [b] This variable was not directly used in the analysis. They were considered as supplementary variables and taken into consideration in the final result.

All variables were reduced to two discriminant functions that together cover 92.2% of the total variance. The first discriminant function defined the position of the samples in the two-dimensional plane as a function of total fiber and total fat, while the second canonical function determined the position of the samples as a function of moisture, Mg and Ca.

Figure 2 shows the scatter diagram whose axes were represented by these two discriminant functions showing the positions of the 44 samples.

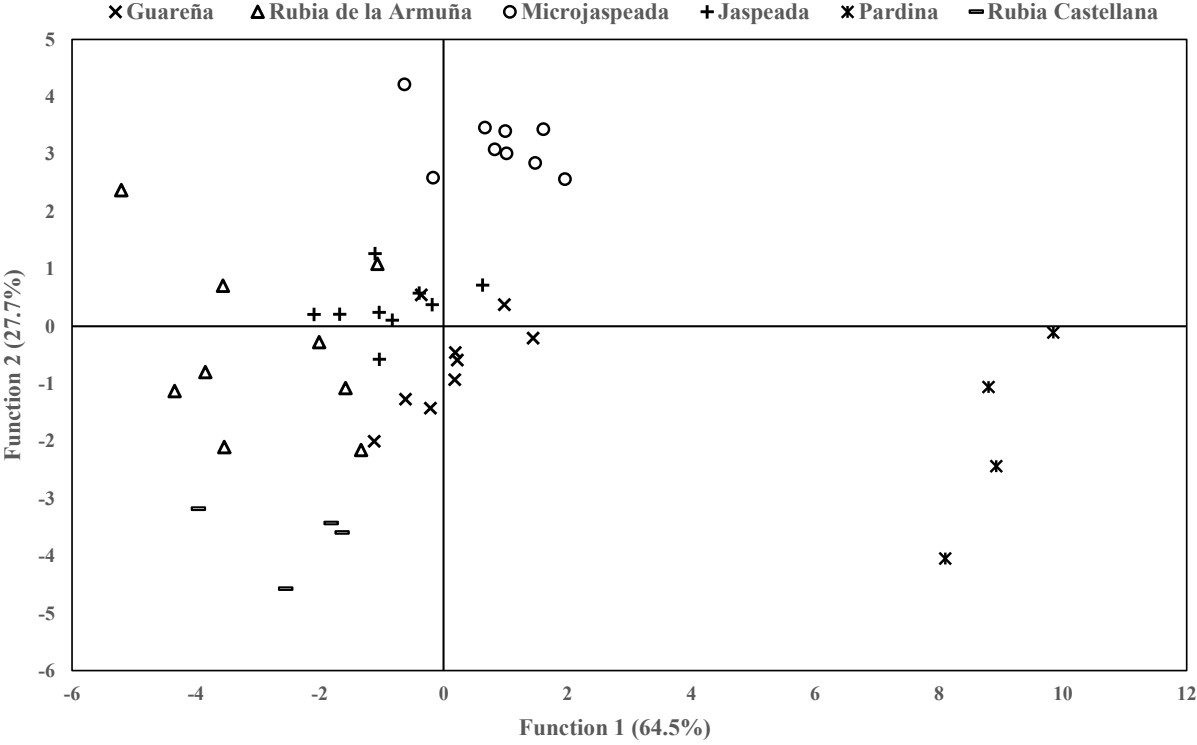

**Figure 2.** Scatter plot that shows the canonical or discriminant coordinates of the analyzed lentil samples. In brackets is the % variance that explains each of the canonical functions.

Figure 3 shows the dendrogram resulting from the cluster analysis (N = 44 samples), from which it was determined that 7.5 was the ideal cut-off distance, since it was the point just before the iterative process showed big gaps between subsequent cluster's distances. This cut-off distance resulted in the formation of four clusters. Furthermore, Table 8 shows the means of each of the variables studied for each cluster.

Clusters 3 and 4 were formed by the samples of the cultivars "Pardina" and "Rubia Castellana", respectively, while clusters 1 and 2 were formed by a combination of samples of cultivars from the PGI "Lenteja de la Armuña". The whole set of samples of the cultivars

"Jaspeada" and "Microjaspeada" was found in cluster 2, together with one sample of cultivar "Rubia de la Armuña" and one sample of cultivar "Guareña". Moreover, all the samples of the cultivar "Rubia de la Armuña", were found in cluster 1, as well as all the samples of the cultivar "Guareña", except for the abovementioned samples of each of these cultivars.

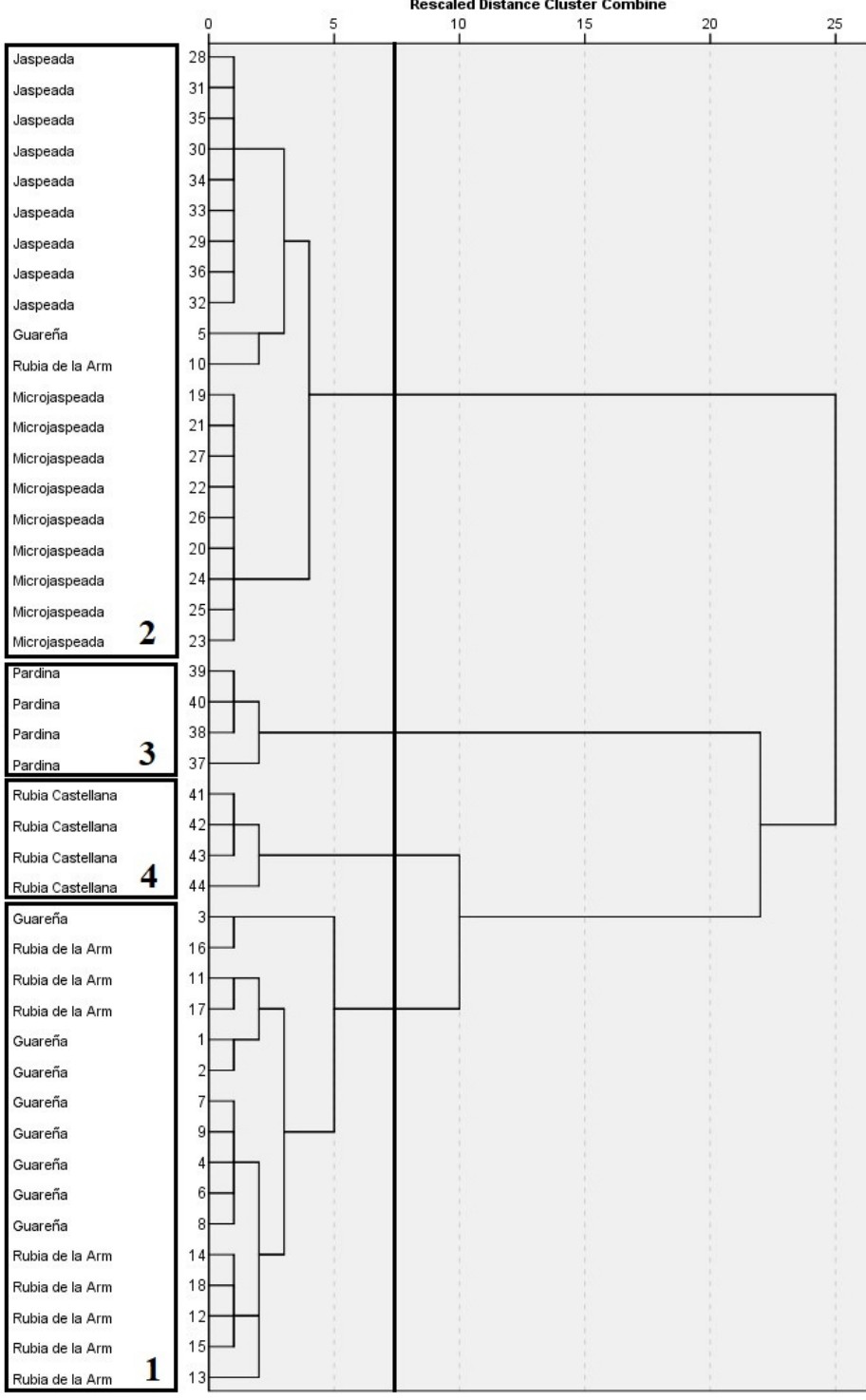

**Figure 3.** Dendrogram obtained from hierarchical cluster analysis with Ward's method. On the left, the lentil samples that make up each of the clusters are grouped together. Each number (1–4) corresponds to each of the formed clusters.

**Table 8.** Means of studied variables for each of the clusters.

| Parameter | Cluster 1 | Cluster 2 | Cluster 3 | Cluster 4 |
|---|---|---|---|---|
| Weigh (g) | 6.49 | 7.21 | 3.59 | 5.91 |
| Size (mm) | 6.67 | 6.66 | 4.43 | 5.87 |
| Colour (%) | 14.87 | 90.00 | 75.41 | 5.86 |
| Protein (%) | 25.02 | 21.51 | 22.43 | 25.02 |
| Carbohydrates (%) | 59.33 | 63.53 | 61.50 | 60.64 |
| Total Fibre (%) | 6.40 | 6.21 | 7.77 | 6.17 |
| Ashes (%) | 2.29 | 2.34 | 2.32 | 2.50 |
| Total fat (%) | 0.64 | 0.73 | 0.90 | 0.54 |
| Moisture (%) | 6.64 | 5.48 | 5.79 | 5.46 |
| Ca (ppm) | 782.81 | 786.63 | 809.19 | 550.19 |
| Fe (ppm) | 135.25 | 126.40 | 98.54 | 106.30 |
| Mg (ppm) | 927.75 | 973.25 | 805.33 | 790.41 |

## 4. Discussion

### 4.1. Morphometric Seed Characteristics

The morphometric results regarding the secondary color pattern distribution are in agreement with the results of Cristóbal et al. [23] and Lázaro et al. [22], and also with the information taken from the specifications of the PGI "Lenteja de la Armuña" (EC Regulation No. 390/2008) and "Tierra de Campos" (EC Regulation No. 510/2006). The intense and frequent black color of the marbled and spotted observed in "Jaspeada" and "Microjaspeada" cultivars could be directly related to their high Mg concentration.

Weight and diameter results are according to those found by Cristóbal et al. [23], who studied the morphological characters of different lentil cultivars from Castilla y León. It was observed that the size of the cultivar "Rubia Castellana" was slightly smaller than that reported by Alonso and Cristóbal [14], furthermore being smaller than the rest of the "Armuña" cultivars. This fact is probably due to the breeding program that has been carried out in the PGI "Lenteja de la Armuña" during the last years, through which larger and larger lentils have been selected. The mean 100-seed weights of both *macrosperma* (6.70 g) and *microsperma* (3.58 g) morphotypes are higher than the means found by Bacchi et al. [27] (5.19 g and 2.88 g, respectively).

### 4.2. Nutritional Seed Quality

Although there were statistical differences regarding moisture, all genotypes showed moisture values well below 15%, which is considered the maximum admissible moisture limit for lentils according to the quality standards of the Spanish PGIs.

Protein values found in this study were lower than those found by Bhatty [35], who reported a protein content between 28% and 32.1% in certain cultivars of Canadian lentils, or those found by Laghetti et al. [28], who gave values ranging from 25.8% in landrace "Onano" to 28.6% in landrace "Ventotene". However, they are similar values to those cited by Moreiras et al. [36], who referred to an average protein content of 23.8%, or to the range reported by Lazzeri et al. [37] who found values between 20.6% and 31.4%. The cultivars "Microjaspeada" and "Jaspeada", together with the cultivar "Pardina", had significantly lower protein values, presumably due to the selection criteria priority on the cultivar "Guareña". According to PGI "Lenteja de la Armuña", the main aim of this selection was to obtain lentil cultivars of high weight and with recognizable characteristics that differentiate them on the market, as it is the spotted or spotted-marbled secondary color pattern they have.

Carbohydrates results are in agreement with those found by Padovani et al. [38] and Faris et al. [7], confirming that carbohydrates represent the major component of lentils, among which starch stands out above the others [39].

As for total fiber content, the fact that "Pardina" cultivar showed the highest value is in agreement with the results obtained by Dueñas et al. [21] in their study on the influence of the cooking process on fiber and polyphenolic compounds content. In general, total fiber results were higher than those found by Bhatty [35], but within the range found by

Moldovan et al. [40] for green lentil cultivars. On the other hand, although there were no statistical differences in ashes content, the range of values, which varied between 2.2% and 2.6%, was lower than that found by Faris et al. [7] and by Moldovan et al. [40].

The fact that the cultivar "Pardina" is the one with the highest fat content is in agreement with Aguilera et al. [41], although these authors report a fat content of 2.8%, much higher than that found in this work (0.9%). Total fat mean content of Spanish cultivars (approx. 0.7%) was considerably lower than that observed for other lentil cultivars by Moreiras et al. [36] which was around 1.8%, by Ryan et al. [42] which was 1.4%, or by Wang & Daun [43] which was 1.3%.

Regarding the main mineral elements, Spanish cultivars' mean contents of Ca, Fe and Mg were remarkably higher than those recorded by Suliburska and Krejpcio [44], who assessed the presence of these components in Polish green lentil cultivars. Furthermore, Ca and Fe levels were in all cases (except Ca in the cultivar "Rubia Castellana") higher than those found by Laghetti et al. [28] in all the Italian landraces. However, the reason why cultivar "Pardina" had the lowest Fe content could be that this cultivar has a higher amount of phytic acid than the other cultivars, which is a polyphenolic compound that inhibits the ICP-OES quantification of Fe due to the formation of chelates, which are catalytically inactive [41,45]. High Mg content of cultivars "Jaspeada" and "Microjaspeada" might be associated with their black-spotted secondary color pattern.

As for the nutritional quality of lentils and other pulses, Boye et al. [46] observed that lentil is the species with the highest protein content in comparison with pea and chickpea cultivars in Canada. Furthermore, lentils are the pulses with the lowest fat and highest fiber content, as supported by the results of El-Adawy et al. [47] in a research study carried out by the Giza Agricultural Research Center (Egypt) comparing the nutritional properties of certain cultivars of beans, peas and lentils. Regarding the main mineral elements, Suliburska and Krejpcio [44] determined in an assessment study of the mineral content of a variety of foodstuffs that green lentil cultivars contain higher amounts of Fe, Zn and Ca than bean or pea cultivars, but lower amounts of Mg.

In short, lentils could be considered a high-quality nutritional food since, in addition to the notable protein content, they provide high carbohydrate content, negligible quantities of fat, and substantial proportions of Ca, Fe, and Mg [48]. The high fiber content in lentils plays an essential role in intestinal health and prevention of diseases from different origins [49]. This fact is a strong point in favor of lentils compared to meat products, which lack fiber. Although the presence in lentil seeds of antinutritional compounds such as protease inhibitors is low in comparison with other pulse species, the low bioavailability of part of the protein and mineral elements due to those compounds can be easily reduced by thermal treatments such as cooking [21,48].

### 4.3. Climatic Conditions Influence on the Studied Parameters

Overall, lentil samples from 2018 season were lighter, smaller, higher in protein, lower in carbohydrate and higher in fiber than lentils from the 2017 season. All of these variations are largely due to the rainfall difference between the two periods, results that are consistent with those found by Lake and Sadras [50].

### 4.4. Stepwise Discriminant and Cluster Analysis

The stepwise discriminant analysis revealed the formation of three clearly differentiated clusters: one, the largest, formed by the samples from PGI "Lenteja de la Armuña" cultivars ("Guareña", "Rubia de la Armuña", "Jaspeada" and "Microjaspeada"); another cluster formed by the four samples of the cultivar "Rubia Castellana" located in the lower part of the diagram and closer to the ordinate axis, which indicates that it is more affected by function 2, and slightly closer to the PGI "Lenteja de la Armuña" cluster too; and a third cluster formed by the four samples of the cultivar "Pardina" located in the right part of the diagram and closer to the abscissa axis, which indicates that it is more affected by function 1. This last cluster's position is strongly conditioned by the large amount of fiber

the cultivar "Pardina" has, and by the direct relationship between the fiber variable and the discriminant function 1. Within the large group of the PGI "Lenteja de la Armuña", the samples corresponding to the cultivar "Rubia de la Armuña" were shifted to the left and this was mainly due to their low total fat content since this variable had a direct relationship with the discriminant function 1. On the other hand, the position of the cultivar "Rubia Castellana" samples was defined not only by their low total fat content, but also by their low Ca and Mg levels. Those mineral elements had a direct relationship with discriminant function 2.

As for the cluster analysis, the means of clusters 3 and 4 corresponded to the means of the cultivars "Pardina" and "Rubia Castellana", respectively. On the other hand, clusters 1 and 2, despite being formed by samples of cultivars belonging to the PGI "Lenteja de la Armuña", showed differential characteristics. Although cluster 1 and 2 lentils were nearly identical in size, the former were lighter lentils, but they had higher protein, total fiber and Fe content. In contrast, cluster 2 was made up of heavier lentils, which also have a higher content of carbohydrates and total fat. Furthermore, cluster 2 higher Ca and Mg content is noteworthy because of the cultivars "Jaspeada" and "Microjaspeada" samples. Thus, both clusters (1 and 2) present well balanced characteristics that refer to high quality groups of lentils. As could be seen, the cluster analysis has grouped genotypically very similar samples, and vice versa. This also suggests that genotypes within the same cluster share some kind of ancestral relationship. These results are in agreement with those obtained by Tyagi and Kahn [51], who investigated genetic diversity in terms of grouping by cluster analysis of 50 lentil genotypes.

## 5. Conclusions

In conclusion, the main Spanish lentil cultivars are high quality food products, as they have remarkable protein (21–25%) and carbohydrates (>59%) content, even higher than those found in any other pulse. Fiber content was higher than expected in "Armuña" and "Rubia Castellana" lentils (6–6.6%), and extremely high in "Pardina" lentil (7.8%). Conversely, very low values were found for fat content (0.5–0.9%). Ca, Fe and Mg levels were notably higher (550–851 ppm, 98–139 ppm and 790–989 ppm, respectively) than those found for other lentil cultivars, especially the high Mg content in the cultivars "Jaspeada" and "Microjaspeada" (>955 ppm). The majority of Spanish lentil cultivars stand out because of their size, weight, protein and mineral elements content. Within these Spanish lentils, cultivars from PGI "Lenteja de la Armuña" are clearly different from the ones of PGI "Tierra de Campos" as well as different from the cultivars of "Rubia Castellana". Overall, the lentil cultivars included in the PGI "Lenteja de la Armuña" have shown better morphometric and nutritional characteristics than the cultivars "Pardina" and "Rubia Castellana".

**Author Contributions:** Conceptualization: M.R.M.-C. and R.P.-S.; Data curation: J.P., M.R.M.-C., R.P.-S., I.R. and A.M.V.-Q.; Formal analysis: J.P., M.R.M.-C. and R.P.-S.; Funding acquisition: M.R.M.-C.; Investigation: J.P., M.R.M.-C., R.P.-S., I.R. and A.M.V.-Q.; Methodology: J.P., M.R.M.-C. and R.P.-S.; Software: J.P.; Project administration: M.R.M.-C.; Resources: R.P.-S.; Supervision: M.R.M.-C. and R.P.-S.; Validation: J.P., I.R. and A.M.V.-Q.; Writing—original draft: J.P.; Writing—review and editing: J.P., M.R.M.-C., R.P.-S., I.R. and A.M.V.-Q. All authors have read and agreed to the published version of the manuscript.

**Funding:** This research was funded by the "Diputación de Salamanca", Project 2015/00239/001.

**Data Availability Statement:** The data presented in this study are available on request from the corresponding author.

**Conflicts of Interest:** The authors declare no conflict of interest.

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
