# Peer review of "Morphometric and Nutritional Characterization of the Main Spanish Lentil Cultivars"

_agriculture, doi:10.3390/agriculture11080741_

Round 1

Reviewer 1 Report

Thank you so much for your work in revising your paper!

Reviewer 2 Report

No further comments!

This manuscript is a resubmission of an earlier submission. The following is a list of the peer review reports and author responses from that submission.

Round 1

Reviewer 1 Report

  • Line 100 According to the paper title, I recommend a reformulation of the sentence: ” …a nutritional and morphometric characterization…” change to “…a morphometric and nutritional characterization…”
  • Line 141 Regarding the measurement of the longitudinal diameter, for the same lentil sample several measurements were carried out? It would be recommended to reduce the impact of measurement errors
  • Line 146 “…Ca, Fe and Mg…” change to “…Ca, Fe and Mg content”
  • Line 149 “15° C” change to “15°C”
  • Line 150 Please specify what is AOAC?
  • Line 151 “540° C” change to “540°C”
  • Line 201-203 Please specify what is the n value in Table 3 and Table 4. The plots?
  • In Table 3 and Table 4 is not correct “Total”. Must be changed to “Mean”
  • Line 235 Why did you exclude the protein and ashes from your analysis?
  • Table 6 “Weigh (gr)” change to “Weigh (g)”
  • Line 336 Taking into account the different meteorological conditions between 2017 and 2018 (Table 1), the values shown in Table 3 and 4 were made separately for each year? Thus, a comparative study is required.

Author Response

Response to Reviewer 1 Comments

We highly appreciate the comments and observations made by the reviewer as they improved the scientific quality of the manuscript.

Point 1. Line 100 According to the paper title, I recommend a reformulation of the sentence: ” …a nutritional and morphometric characterization…” change to “…a morphometric and nutritional characterization…”.

Response 1. Revised as requested.

Line 100: Therefore, the aim of the present study is to carry out a morphometric and nutritional characterization of the main Spanish lentil cultivars.

Point 2. Line 141 Regarding the measurement of the longitudinal diameter, for the same lentil sample several measurements were carried out? It would be recommended to reduce the impact of measurement errors.

Response 2. The reviewer is right. Three measurements for the same lentil sample were carried out. The mean of these three measurements was calculated for each of the 70 lentils of each cultivar assessed. Subsequently, the mean of the 70 lentils was averaged for each cultivar. This information has been added to the manuscript.

Line 146: As for the longitudinal diameter, three measurements for the same lentil sample were performed. The mean of these three measurements was calculated for each of the 70 lentils of each cultivar assessed. Subsequently, those 70 mean results were averaged for each cultivar.

Point 3. Line 146 “…Ca, Fe and Mg…” change to “…Ca, Fe and Mg content”.

Response 3. Revised as requested.

Line 145: Nutritional quality was quantified by analyzing the following parameters: moisture, Ca, Fe and Mg content…

Point 4. Line 149 “15° C” change to “15°C”.

Response 4. Revised as requested.

Line 148: The samples were ground with their skin in Foss KnifetecTM 1095 mill (FOSS, Hilleroed, Denmark), with temperature control at 15°C.

Point 5. Line 150 Please specify what is AOAC?

Response 5. AOAC means Association of Official Analytical Collaboration (AOAC) International. It has been added to the text.

Line 150: For moisture calculations, samples were dried following the Association of Official Analytical Collaboration (AOAC) International procedures [32].

Point 6. Line 151 “540° C” change to “540°C”.

Response 6. Revised as requested.

Line 151: The ash content was determined by quantifying the residue after combustion of the dry sample in a muffle furnace at 540°C for 6 hours under conditions corresponding to the gravimetric method.

Point 7. Line 201-203 Please specify what is the n value in Table 3 and Table 4. The plots?

Response 7. Thank you for the comment. The reviewer is right, n refers to the number of plots. It has been added to Table 3 and 4 footnotes.

Point 8. In Table 3 and Table 4 is not correct “Total”. Must be changed to “Mean”

Response 8. Thank you for the correction. “Total” has been changed to “Mean” in Tables 3 and 4.

Point 9. Line 235 Why did you exclude the protein and ashes from your analysis?

Response 9. The authors did not exclude the variables "protein" and "ash" from the stepwise discriminant analysis. The algorithm underlying this statistical procedure determined at a certain point that these two variables provided irrelevant information that distorted the discrimination results, and therefore it decided to exclude them from the process. This is one of the great advantages of the stepwise algorithmic methodology, since it compares the combinations variable by variable in search of the most representative ones.

Point 10. Table 6 “Weigh (gr)” change to “Weigh (g)”.

Response 10. Revised as requested.

Point 11. Line 336 Taking into account the different meteorological conditions between 2017 and 2018 (Table 1), the values shown in Table 3 and 4 were made separately for each year? Thus, a comparative study is required.

Response 11. Thank you very much for the correction. The reviewer is completely right. The comparative study between growing season was already done, but the results were not shown in the manuscript. In order to fulfil this requirement, the next data and information have been added to the main text.

Line 183: This procedure was also used to study significant differences between growing seasons.

Line 201: It should be noted that the results shown below correspond to the average data of the two growing seasons (2017 and 2018). In the studies dealing with plant material characterization, it is essential to assess different growing seasons to take into account climatic conditions influences, but the results are usually expressed as the average data of the seasons involved, as in the case of the study carried out by Bacchi et al [26]. However, the comparative study between growing seasons was carried out and the synthetic results are shown in table 6.

Line 251: 3.3. Climatic conditions influence on the studied parameters

As shown in Table 1, 2018 growing season was hotter and considerably drier than the 2017 season. Overall, this variation significantly (p<0.05) affected seed size and weight, as well as protein, carbohydrate and fiber content. These results are shown in Table 6.

Table 6. Growing season’s differences in the morphometric and nutritional parameters of the studied cultivars.

Cultivar

Morphometric parameters

Nutritional parameters

FO

MC

SCP

SCPD

WE

DI

MO

PR

CH

TFi

AS

TFa

Ca

Fe

Mg

Guareña

ND

ND

ND

ND

*

*

ND

*

*

*

ND

ND

*

ND

ND

Rubia de la Armuña

ND

ND

ND

ND

*

*

ND

*

*

*

ND

ND

ND

ND

ND

Microjaspeada

ND

ND

ND

ND

*

*

ND

*

*

*

ND

ND

ND

ND

*

Jaspeada

ND

ND

ND

ND

*

*

ND

*

*

*

ND

ND

ND

ND

ND

Pardina

ND

ND

ND

ND

*

*

ND

*

*

*

ND

ND

ND

*

ND

Rubia Castellana

ND

ND

ND

ND

*

*

ND

*

*

*

*

ND

ND

ND

ND

* Significant differences in the row p<0.05, ND: no significant differences in the row p>0.05.

FO: form, MC: main color, SCP: secondary color pattern, SCPD: secondary color pattern distribution (%), WE: weight (g), DI: diameter (mm), MO: moisture (%), PR: protein (%), CH: carbohydrates (%), TFi: total fiber (%), AS: ashes (%), TFa: total fat (%), Ca: calcium (ppm), Fe: iron (ppm), Mg: magnesium (ppm).

Reviewer 2 Report

This is an interesting study reg nutrient content and morphology of Spanish lentils.

Please pay attention to the following:

Please provide information in the abstract if nutrient content is on dry matter or fresh weight basis. And provide information on fiber content. Give range on fat and micronutrient content, very "high" and "low" is too unspecific. Spell out PGI/explain.

The introduction requires attention and a language check by a professional.
In several text passages, clarification is need when referring with the terms "them", "their", see attached SOME examples.

Minor mistakes and unclear passages might need amendments. Tannins are polyphenols, and when mentioning nutritional advantages of pulses it is good to clarify that the reference 16 is referring to in vitro antiox capacity.

The level of information given is uneven, e.g. lines 50 ff regarding nutrient content is less detailed as  information regarding cultivation/cultivars, lines 63-65 and ff.

In the section material and methods, I had preferred a clear table with information reg samples, cultivars, subsp, and number of plots. How did you handle the material from the two seasons? Pl clarify.
This is relevant to understand the statistical results.

Pl clarify how samples were dried (l 142).

Ref 31 is not accessible, a brief description of procedures is required.

Lines 149 ff, please supply for methodology (ref 32),  briefly information reg temp (105C?, time?) and if drying was carried out to constant weight.

How was the colour established? And the relative distribution pattern? Pl clarify in method section.

Results:

Pl clarify (n= xx) in tables' legends.

In table 3, I expected data from colorimetric analysis with respect to L, a and b, not phrases punk/green reg main colour. I also wonder how the secondary color pattern distribution could be established to an accuracy of two decimals. Pl clarify.

Table 4: I query that crude protein and other proximates can be established to a percentage with the accuracy of two decimals. This is also valid for carbohydrate content by difference. Please correct. I neither consider it relevant with two decimals for data on mineral content.

Fig 3 shows data from n=44. This is unclear with respect to information given under material and methods, 44 plots per season.

Pl clarify why ash and protein were excluded from stat analysis, line 235 ff. No clear statement/sentence.

Discussion:

Lines 281ff. When comparing nutrient content (as percentage), are data from others also based on fresh weight (as harvested and dried) and not given as dry matter? This  would also be relevant if data (expressed as % fresh weight) are compared from samples where the water content is close to 15% or over.

Line 316 ff, here it would be only relevant to explain why phytic acid would affect ICP-OES quantification of Fe (rather than bioavailability).

Lines 328 ff. I disagree! The protein quality and digestibility is higher in meat, as well as the bioavailability of minerals (esp as Fe is available as heme-Fe). Fe absorption from meat products  is furthermore increase by MFP factor. Reasoning reg fat content (and i- n theory - fat quality would - be correct (but as you did not determine fatty acid profile data of the study do not support statements regarding fat quality). 
Further advantage of lentils as compared to meat products? (Think of fiber).

Lines 337 ff. Where do I find the data supporting this statement? Pl clarify in the material and method and result section which data derive from the seeds from the different harvest years.

The discussion section 4.4 reg cluster analysis is descriptive and no real discussion. Streamline, add appropriate info into the results chapter. Rephrase this chapter.

Conclusions:

Rephrase. What is "high quality foodsstuff"? Unclear phrasing in entire paragraph, see comments. Rephrase more precise. What is the conclusion?

Author Response

Response to Reviewer 2 Comments

We highly appreciate the comments and observations made by the reviewer as they improved the scientific quality of the manuscript.

Point 1. Please provide information in the abstract if nutrient content is on dry matter or fresh weight basis. And provide information on fiber content. Give range on fat and micronutrient content, very "high" and "low" is too unspecific. Spell out PGI/explain.

Response 1. Thank you for your corrections. The abstract has been modified.

Nowadays, there is a growing demand for vegetal high-quality protein food products, such as pulses and lentils in particular. However, there is no scientific evidence on the nutritional and morphometric characterization of the main lentil cultivars in the Western Mediterranean area. For this reason, the aim of this work is to carry out a morphometric and nutritional characterization of the main Spanish lentil cultivars. Nutrient content assessment was performed on dry matter. The results showed that all studied cultivars are large and heavy lentils, except for the cultivar “Pardina”. They have high protein levels, ranging from 21% to 25%, which is higher than those found in any other pulse, as well as a high carbohydrate content, greater than 59% in all cases. Fiber content was higher than expected in “Armuña” and “Rubia Castellana” cultivars, ranging from 6% to 6.6%, and exceptionally high in the case of the cultivar “Pardina”, which reached 7.8%. Conversely, very low values were found for fat content, varying between 0.5% and 0.9%. Ca, Fe and Mg levels were remarkably higher (from 550ppm to 851ppm, from 98ppm to 139ppm and from 790ppm to 989ppm, respectively) than those found for other lentil cultivars, especially the high Mg content in the cultivars “Jaspeada” and “Microjaspeada”, both above 955 ppm. Clear differentiation was found between the cultivars “Rubia Castellana”, “Pardina” and those included in the Protected Geographical Indication (PGI) “Lenteja de la Armuña”. Overall, lentil cultivars included in the PGI “Lenteja de la Armuña” showed better morphometric and nutritional characteristics than cultivars “Pardina” or “Rubia Castellana”. The nutritional parameters of Spanish lentils are in accordance with the values of the Turkish’s and Italians ones.

Point 2. The introduction requires attention and a language check by a professional. In several text passages, clarification is need when referring with the terms "them", "their", see attached SOME examples.

Response 2. Revised as requested.

Point 3. Minor mistakes and unclear passages might need amendments. Tannins are polyphenols, and when mentioning nutritional advantages of pulses it is good to clarify that the reference 16 is referring to in vitro antiox capacity.

Response 3. Revised as requested.

Line 58: Moreover, lentils are high in fiber [15], water-soluble vitamins, essential minerals and numerous phenolic compounds such as tannins, which are correlated with high in vitro antioxidant capacity [16].

Point 4. The level of information given is uneven, e.g. lines 50 ff regarding nutrient content is less detailed as information regarding cultivation/cultivars, lines 63-65 and ff.

Response 4. Agree with the reviewer. Further information on the nutritional characteristics of lentils has been added.

Line 61: Lentils are considered as soft seed-coated pulses requiring a shorter cooking time, reason why the usually have smaller nutrient losses than those with a hard seed coat [17].

Reference: 17. Satya, S., Kaushik, G., Naik, S.N. Processing of food legumes: a boon to human nutrition. Med. J. Nutrit. Metab. 2010, 3, 183-195, doi: 10.1007/s12349-010-0017-8.

Point 5. In the section material and methods, I had preferred a clear table with information reg samples, cultivars, subsp, and number of plots. How did you handle the material from the two seasons? Pl clarify. This is relevant to understand the statistical results.

Response 5. Thank you for your recommendation. A new table (Table 3) showing samples, cultivars, subsp. and plots information has been added.

Table 3. Identification of the studied cultivar and number of experimental plots used.

PGI

Cultivar

Subsp.

No. Plots

“Lenteja de la Armuña”

“Guareña”

Macrosperma

9

“Lenteja de la Armuña”

“Rubia de la Armuña”

Macrosperma

9

“Lenteja de la Armuña”

“Jaspeada”

Macrosperma

9

“Lenteja de la Armuña”

“Microjaspeada”

Macrosperma

9

“Tierra de Campos”

“Pardina”

Microsperma

4

-

“Rubia Castellana”

Macrosperma

4

Regarding the two growing seasons, the results presented in Table 3 and 4 refer to average data of both seasons. Nevertheless, the comparative study between growing season was already done, but the results were not shown in the manuscript. In order to fulfil this requirement, the next data and information have been added to the main text.

Line 183: This procedure was also used to study significant differences between growing seasons.

Line 201: It should be noted that the results shown below correspond to the average data of the two growing seasons (2017 and 2018). In the studies dealing with plant material characterization, it is essential to assess different growing seasons to take into account climatic conditions influences, but the results are usually expressed as the average data of the seasons involved, as in the case of the study carried out by Bacchi et al [26]. However, the comparative study between growing seasons was carried out and the synthetic results are shown in table 6.

Line 251: 3.3. Climatic conditions influence on the studied parameters

As shown in Table 1, 2018 growing season was hotter and considerably drier than the 2017 season. Overall, this variation significantly (p<0.05) affected seed size and weight, as well as protein, carbohydrate and fiber content. These results are shown in Table 6.

Table 6. Growing season’s differences in the morphometric and nutritional parameters of the studied cultivars.

Cultivar

Morphometric parameters

Nutritional parameters

FO

MC

SCP

SCPD

WE

DI

MO

PR

CH

TFi

AS

TFa

Ca

Fe

Mg

Guareña

ND

ND

ND

ND

*

*

ND

*

*

*

ND

ND

*

ND

ND

Rubia de la Armuña

ND

ND

ND

ND

*

*

ND

*

*

*

ND

ND

ND

ND

ND

Microjaspeada

ND

ND

ND

ND

*

*

ND

*

*

*

ND

ND

ND

ND

*

Jaspeada

ND

ND

ND

ND

*

*

ND

*

*

*

ND

ND

ND

ND

ND

Pardina

ND

ND

ND

ND

*

*

ND

*

*

*

ND

ND

ND

*

ND

Rubia Castellana

ND

ND

ND

ND

*

*

ND

*

*

*

*

ND

ND

ND

ND

* Significant differences in the row p<0.05, ND: no significant differences in the row p>0.05.

FO: form, MC: main color, SCP: secondary color pattern, SCPD: secondary color pattern distribution (%), WE: weight (g), DI: diameter (mm), MO: moisture (%), PR: protein (%), CH: carbohydrates (%), TFi: total fiber (%), AS: ashes (%), TFa: total fat (%), Ca: calcium (ppm), Fe: iron (ppm), Mg: magnesium (ppm).

Point 6. Pl clarify how samples were dried (l 142).

Response 6. Revised as requested. Information about drying procedure has been added.

Line 153: For moisture calculations, samples were dried following the Association of Official Analytical Collaboration (AOAC) International procedures [32], i.e., samples were dried in a conventional oven at 100 °C during 3 h until constant weight.

Point 7. Ref 31 is not accessible, a brief description of procedures is required.

Response 7. Ref. 31 refers to UPOV Guidelines and it is accessible in the next link:

On page 15 of the abovementioned document is available all the information regarding the morphometric parameters measured in this study. Indeed, some information about the procedure is already presented in the text.

Point 8. Lines 149 ff, please supply for methodology (ref 32), briefly information reg temp (105C?, time?) and if drying was carried out to constant weight.

Response 8. Revised as requested. See Response 6.

Point 9. How was the colour established? And the relative distribution pattern? Pl clarify in method section.

Response 9. The color was established by following the abovementioned UPOV guidelines. Some screenshots about it are presented here.

Line 142: The distribution of the secondary color pattern on the lentil surface (%) was estimated by dividing the lentil into four quadrants in a Petri dish, and then relating the number of mottles in each quadrant by its surface area, and averaging the four quadrants results to obtain the secondary pattern's final percentage of coverage.

Point 10. Pl clarify (n= xx) in tables' legends.

Response 10. Revised as requested. “n” means number of plots. It has been added to table’s legends.

Point 11. In table 3, I expected data from colorimetric analysis with respect to L, a and b, not phrases punk/green reg main colour. I also wonder how the secondary color pattern distribution could be established to an accuracy of two decimals. Pl clarify.

Response 11. Although we agree with the reviewer that this analysis should provide relevant information, according to the International Union for the Protection of New Varieties of Plants (UPOV) guidelines, colorimetric analysis is not mandatory for plant material characterization.

The two decimal digits do not indicate the precision of the method, but are the result of averaging the results. As this can lead to confusion, we believe it is convenient to set all results (except for Weight, Diameter and Moisture) to one decimal place. Thank you very much for your contribution.

Point 12. Table 4: I query that crude protein and other proximates can be established to a percentage with the accuracy of two decimals. This is also valid for carbohydrate content by difference. Please correct. I neither consider it relevant with two decimals for data on mineral content.

Response 12. Revised as requested. See Response 11.

Point 13. Fig 3 shows data from n=44. This is unclear with respect to information given under material and methods, 44 plots per season.

Response 13. The reviewer is right. Figure 3 shows data from n=44 observations which resulted from the average value for both growing seasons. In other words, the 44 observations correspond to data showed in Tables 3 and 4.

Point 14. Pl clarify why ash and protein were excluded from stat analysis, line 235 ff. No clear statement/sentence.

Response 14. The authors did not exclude the variables "protein" and "ash" from the stepwise discriminant analysis. The algorithm underlying this statistical procedure determined at a certain point that these two variables provided irrelevant information that distorted the discrimination results, and therefore it decided to transform those variables into supplementary information, that is, they were indirectly considered in the final result. This is one of the great advantages of the stepwise algorithmic methodology, since it compares the combinations variable by variable in search of the most representative ones.

Table footnote explanation has been modified: This variable was not directly used in the analysis. They were considered as supplementary variables and taking into consideration in the final result.

Point 15. Lines 281ff. When comparing nutrient content (as percentage), are data from others also based on fresh weight (as harvested and dried) and not given as dry matter? This would also be relevant if data (expressed as % fresh weight) are compared from samples where the water content is close to 15% or over.

Response 15. All nutrient parameters were calculated on dry matter. Moisture was the first analysis that was performed (after the morphometric assessment). Consequently, the rest of the nutritional characteristics were determined on dry matter.

Point 16. Line 316 ff, here it would be only relevant to explain why phytic acid would affect ICP-OES quantification of Fe (rather than bioavailability).

Response 16. Revised as requested.

Line 318: However, the reason why cultivar “Pardina” had the lowest Fe content could be that this cultivar has a higher amount of phytic acid than the other cultivars, which is a polyphenolic compound that inhibits the ICP-OES quantification of Fe due to the formation of chelates, which are catalytically inactive [42,46].

Point 17. Lines 328 ff. I disagree! The protein quality and digestibility is higher in meat, as well as the bioavailability of minerals (esp as Fe is available as heme-Fe). Fe absorption from meat products is furthermore increase by MFP factor. Reasoning reg fat content (and i- n theory - fat quality would - be correct (but as you did not determine fatty acid profile data of the study do not support statements regarding fat quality). Further advantage of lentils as compared to meat products? (Think of fiber).

Response 17. The reviewer is completely right. This paragraph has been modified. Thank you very much for the recommendation.

Line 357: In short, lentils could be considered a high-quality nutritional food since, in addition to the notable protein content, they provide high carbohydrate content, negligible quantities of fat, and substantial proportions of Ca, Fe, and Mg [49]. The high fiber content in lentils plays an essential role in intestinal health and prevention of diseases from different origins [50]. This fact is a strong point in favor of lentils compared to meat products, which lack fiber.

Reference: Phillips, G.O. Dietary fibre: A chemical category or a health ingredient? Bioac. Carboh. And Diet. Fibre. 2013, 1, 3-9, doi: 10.1016/j.bcdf.2012.12.001.

Point 18. Lines 337 ff. Where do I find the data supporting this statement? Pl clarify in the material and method and result section which data derive from the seeds from the different harvest years.

Response 18. Revised as requested. See Response 5.

Point 19. The discussion section 4.4 reg cluster analysis is descriptive and no real discussion. Streamline, add appropriate info into the results chapter. Rephrase this chapter.

Response 19. Revised as requested. The first part of this section has been moved to the results chapter. The second part has been modified.

Line 397: As for the cluster analysis, the means of clusters 3 and 4 corresponded to the means of the cultivars “Pardina” and “Rubia Castellana”, respectively. On the other hand, clusters 1 and 2, despite being formed by samples of cultivars belonging to the PGI “Lenteja de la Armuña”, showed differential characteristics. Although cluster 1 and 2 lentils were nearly identical in size, the former were lighter lentils, but they had higher protein, total fiber and Fe content. In contrast, cluster 2 was made up of heavier lentils, which also have a higher content of carbohydrates and total fat. Furthermore, cluster 2 higher Ca and Mg content is noteworthy because of the cultivars “Jaspeada” and “Microjaspeada” samples. Thus, both clusters (1 and 2) present well balanced characteristics that refer to high quality groups of lentils. As could be seen, the cluster analysis has grouped genotypically very similar samples, and vice versa.  This also suggests that genotypes within the same cluster share some kind of ancestral relationship. These results are in agreement with those obtained by Tyagi and Kahn [52], who investigated genetic diversity in terms of grouping by cluster analysis of 50 lentil genotypes.

Reference: Tyagi, Sd. and Khan, Mh. Genetic divergence in Lentil. Afr. Crop Sci. J. 2011, 18, 2, 69-74, doi: 10.4314/acsj.v18i2.65798.

Point 20. Rephrase. What is "high quality foodsstuff"? Unclear phrasing in entire paragraph, see comments. Rephrase more precise. What is the conclusion?

Response 20. Revised as requested. Conclusions have been modified.

Line 408: In conclusion, the main Spanish lentil cultivars are high quality food products, as they have remarkable protein (21%-25%) and carbohydrates (> 59%) content, even higher than those found in any other pulse. Fiber content was higher than expected in “Armuña” and “Rubia Castellana” lentils (6%-6.6%), and extremely high in “Pardina” lentil (7.8%). Conversely, very low values were found for fat content (0.5%-0.9%). Ca, Fe and Mg levels were notably higher (550ppm-851ppm, 98ppm-139ppm and 790ppm-989ppm, respectively) than those found for other lentil cultivars, especially the high Mg content in the cultivars “Jaspeada” and “Microjaspeada” (> 955 ppm). The majority of Spanish lentil cultivars stand out because of their size, weight, protein and mineral elements content. Within these Spanish lentils, cultivars from PGI “Lenteja de la Armuña” are clearly different from the ones of PGI “Tierra de Campos” as well as different from the cultivars of “Rubia Castellana”. Overall, the lentil cultivars included in the PGI “Lenteja de la Armuña” have shown better morphometric and nutritional characteristics than the cultivars “Pardina” and “Rubia Castellana”.
